# Predictors of Mortality in Early Neonatal Sepsis: A Single-Center Experience

**DOI:** 10.3390/medicina59030604

**Published:** 2023-03-18

**Authors:** Marija Jovičić, Miloš N. Milosavljević, Marko Folić, Radiša Pavlović, Slobodan M. Janković

**Affiliations:** 1Institute of Neonatology, 11000 Belgrade, Serbia; mexicomara@gmail.com; 2Department of Pharmacology and Toxicology, Faculty of Medical Sciences, University of Kragujevac, 34000 Kragujevac, Serbia; slobnera@gmail.com; 3Department of Pharmacy, Faculty of Medical Sciences, University of Kragujevac, 34000 Kragujevac, Serbia; markof@medf.kg.ac.rs (M.F.); rpavlovic@medf.kg.ac.rs (R.P.); 4Clinical Pharmacology Department, University Clinical Centre Kragujevac, 34000 Kragujevac, Serbia

**Keywords:** neonatal sepsis, newborn, neonatology, pediatricians, case-control studies, logistic models

## Abstract

*Background and Objectives*: Early neonatal sepsis is associated with a significant mortality rate despite modern treatment strategies. Our aim was to identify risk factors contributing to the occurrence of death in newborns with early neonatal sepsis. *Materials and Methods*: We conducted a retrospective cross-sectional study that included newborns with early sepsis who received care in the intensive and semi-intensive care units at the Institute of Neonatology, Belgrade, Serbia. Newborns with early neonatal sepsis who died comprised the case group, whereas those who survived made up the control group. The diagnostic and therapeutic approach to the septic condition was carried out independently of this study, according to valid hospital protocols and current good practice guidelines. The influence of a large number of variables on the examined dichotomous outcome, as well as the mutual interaction of potential predictor variables, was examined by binary logistic regression. *Results*: The study included 133 pregnant women and 136 newborns with early neonatal sepsis, of which 51 (37.5%) died, while the remaining 85 newborns (62.5%) survived. Newborns who died had a statistically significantly lower birth weight compared to those who survived (882.8 ± 372.2 g vs. 1660.9 ± 721.1 g, *p* = 0.000). Additionally, compared to newborns who survived, among the deceased neonates there was a significantly higher proportion of extremely preterm newborns (74.5% vs. 22.4%, *p* = 0.000). The following risk factors for the occurrence of death in early neonatal sepsis were identified: low birth weight, sepsis caused by gram-negative bacteria, and the use of double-inotropic therapy and erythrocyte transfusion during the first week. *Conclusions*: Pediatricians should pay special attention to infants with early neonatal sepsis in whom any of the identified risk factors are present in order to prevent a fatal outcome.

## 1. Introduction

Early neonatal sepsis, which is commonly described as sepsis that occurs during the first 72 h after delivery, is still a significant factor in neonatal mortality [1]. The causative agents of the infection are microorganisms that colonize the maternal genitourinary tract. Group B Streptococcus (GBS) remains the most common cause of early neonatal sepsis apparent in term infants, while Escherichia coli is the most common cause of early sepsis in premature infants [1,2]. Some of the risk factors for early neonatal sepsis identified so far are: prematurity, low birth weight, congenital anomalies, low Apgar score, invasive procedures during pregnancy, virulence of microorganisms, premature rupture of fetal membranes, a sibling with history of an invasive GBS infection, chorioamnionitis, elevated maternal body temperature, poor prenatal care, low socioeconomic status, abuse of psychoactive substances, and black race [2,3,4,5,6,7].

The gold standard for the diagnosis of sepsis is a positive blood culture finding, where care must be taken with the volume of blood drawn, as an insufficient amount of blood drawn for blood culture cannot detect low levels of bacteremia [8]. The amount of blood taken for a blood culture is particularly important for diagnosing early neonatal sepsis. Thus, it is considered that the amount of blood in 1 milliliter provides a sensitivity of about 90% in diagnosing early neonatal sepsis [9]. The timely, prompt, and adequate administration of antibiotics in neonatal sepsis is essential. However, the prolonged use of antibiotics in infants who are not suffering from sepsis delays the identification of other conditions that are similar to sepsis, and is associated with the emergence of bacterial resistance, fungal infections, late-onset sepsis, and necrotizing enterocolitis [10,11,12,13]. Hence, clinicians are continually searching for new markers that might provide high selectivity and specificity for the diagnosis of early neonatal sepsis, such as inflammatory markers, cytokines, different hematological parameters, and indicators for determining nutritional status [14,15,16,17,18,19,20,21]. 

Early neonatal sepsis is associated with a significant mortality rate of up to 54% [1], despite modern treatment strategies. Therefore, our aim was to identify risk factors contributing to the occurrence of death in newborns with early neonatal sepsis.

## 2. Materials and Methods

### 2.1. Study Design and Study Population

This study was a retrospective cross-sectional study. The study included newborns with early sepsis who received care in the intensive and semi-intensive care units at the Institute of Neonatology, Belgrade, Serbia, between 2013 and 2015. The study included two groups of participants: a case group composed of newborns with early sepsis who died, and a control group made up of newborns with sepsis, and other characteristics similar those of the cases, who survived hospital care and were released as cured. The diagnostic and therapeutic approach to the septic condition was carried out independently of this study, according to valid hospital protocols and current good practice guidelines related to the central topic of this research. 

Newborns who met the following inclusion criteria participated in the study: (1) sepsis identified within the first 72 h of life and (2) positive blood culture and/or cerebrospinal fluid (CSF) results. The criteria for excluding the patient from the clinical trial were: (1) the presence of major congenital anomalies; (2) incomplete medical documentation; and (3) violation of the study protocol. The study used a so-called “convenient” sample, and in order to reduce the influence of the researcher’s bias, the sample was consecutive, i.e., it integrated all newborns who met the inclusion criteria and were treated in the period from 1 January 2013 to 1 January 2015.

### 2.2. Variables Measured in the Study

The death outcome in newborns with early neonatal sepsis was determined by reviewing their medical records. In the domain of maternal factors with a potential significant influence on the outcome of interest, the following variables were analyzed: the method of conception (natural or in vitro fertilization), method of delivery (vaginal or caesarean section), singleton/multiple pregnancy, and the mother’s history of previous births, history of previous miscarriages, prenatal conditions/diseases (chronic diseases, i.e., diseases and conditions related to pregnancy), age, and available microbiological analyses (cervical and vaginal smear findings, urine culture findings). The effect of the following newborn-related variables was examined on the dependent variable: gender, body weight at birth, gestational age at birth, Apgar score in the 1st minute, body temperature on admission to Institute of Neonatology, verified microbiological cause of early neonatal sepsis with antibiogram, biochemical and clinical parameters (average value of blood pH on the 1st day, average value of base excess on the 1st day, average value of the fraction of inspired oxygen during the 1st day, average value of systolic blood pressure on the 1st day, average value of diastolic blood pressure on the 1st day, average value of mean blood pressure on the 1st day, leukocytes on the 1st day, thrombocytes on the 1st day, hemoglobin on the 1st day, albumins on the 1st day, urea on the 3rd day, creatinin on the 3rd day, bilirubin on the 2nd day, the presence of convulsions during the 1st week, the presence of pneumothorax/pneumomediastinum/pulmonary interstitial emphysema during the 1st week), conducted therapeutic measures (resuscitation measures, the use of ibuprofen for the treatment of ductus arteriousus, the use of surfactant), the use of double-inotropic therapy, the use of phototherapy, initial antibiotic therapy, protein intake during the 1st day, glucose intake during the 1st day, fluid intake during the 1st day, erythrocyte transfusion during the 1st week, thrombocyte transfusion during the 1st week, and plasma transfusion during the 1st week.

### 2.3. Statistical Data Analysis

Statistical data analysis was performed using the SPSS software package, version 18 (SPSS Inc., Chicago, IL, USA). The methods of descriptive statistics were used to show the basic characteristics of the newborns. Mean values ± standard deviation with range (minimum–maximum) were used for continuous variables, and relative frequency was used for categorical variables. The distribution of numerical data was tested for normality using the Kolmogorov–Smirnov and Shapiro–Wilk tests. Student’s t-test for independent samples determined the existence of a statistically significant difference between the compared groups in the values of continuous variables, provided that the distribution is normal; otherwise, the nonparametric alternative Mann–Whitney U test was used. Comparisons between categorical variables were made using the χ^2^ test (or Fisher’s exact probability test for the low frequency of certain categories). If the probability of the null hypothesis was lower than 5% (*p* = 0.05), the difference was considered statistically significant. The influence of a large number of variables on the examined dichotomous outcome, as well as the mutual interaction of potential predictor variables, was examined by binary logistic regression, and the results are presented in the form of a raw and adjusted odds ratio (OR) with the associated 95% confidence interval.

## 3. Results

### 3.1. Characteristics of Pregnant Women

The study included 133 pregnant women with a mean (SD) age of 30.5 (5.7) years. The basic clinical characteristics of the pregnant women are shown in Table 1. 

### 3.2. Characteristics of Newborns

In total, the study included 136 newborns with early neonatal sepsis (both twins in three mothers developed early neonatal sepsis), of which 51 (37.5%) died (cases), while the remaining 85 newborns (62.5%) survived (controls). The basic characteristics of the newborns at birth are shown in Table 2. Compared to the control group, the newborns who died had statistically significantly lower values of birth weight and Apgar score in the first minute. Additionally, among the cases, there was a significantly higher proportion of extremely and very preterm newborns compared to the control group.

All newborns included in this study were diagnosed with early neonatal sepsis and all had a positive blood culture. The most important biochemical and clinical characteristics of the newborns by group are shown in Table 3, while the presentation of the applied therapeutic measures is given in Table 4. 

### 3.3. Risk Factors for Fatal Outcome in Early Neonatal Sepsis

The results of both univariate and multivariate binary logistic regression from the last step with satisfactory goodness of fit (Cox and Snell R square 0.546, Nagelkerke R2 0.746, Hosmer–Lemeshow Chi-square 9.268, df = 8, *p* = 0.320, overall model accuracy of 89.6%) with adjustment for potential confounders are shown in Table 5. The variables entered in the multivariate analysis were: birth weight, average value of blood pH on the first day, body temperature on admission, double-inotropic therapy, convulsions during the first week, erythrocyte transfusion during the first week, plasma transfusion during the first week, and blood culture. After adjustment for potential confounders and other independent variables, the following risk factors for the occurrence of death in early neonatal sepsis were identified: low birth weight, sepsis caused by gram-negative bacteria (GNB), and the use of double-inotropic therapy and erythrocyte transfusion during the first week.

## 4. Discussion

Sepsis in the pediatric population has been extensively studied in recent years. With more than 1.2 million cases per year [22] and 25% of fatal outcomes worldwide [23], it remains among the leading causes of childhood mortality. It is important to acknowledge that 84% of infant deaths due to sepsis are preventable [24], and the understanding of its risk profile may contribute to the development of risk-prediction models aimed at improving the overall survival of this group of patients. 

The main objective of the present study was to identify potential predictors for the fatal outcome of sepsis in newborns. Our results showed birth weight to be a protective factor. The incidence of a fatal outcome increases as birth weight decreases. In a systematic review with a meta-analysis of 240 studies, the highest case fatality rate of 24% was observed in the group of patients with very low birth weight (<1500 g), while the incidence rate was the lowest, at 15%, in infants with a birth weight ≥2500 g [24]. The average birth weight in our case group was 882.8 ± 372.2, indicating the possibility of a high mortality rate. In addition to the incidence rate, one of the studies analyzed in the systematic review noted a more than sixfold-increased risk of a fatal outcome in newborns with a low birth weight [25]. Another major risk factor in the same study was the time of delivery, showing a strong association between prematurity and severe sepsis onset. Newborns who are prematurely born have a relatively immature immune system, are more susceptible to infections, and are at higher risk for mortality [26]. Given that 93 percent of all of our patients were preterm and that we had a higher number of extremely preterm and very preterm newborns compared to the control group, the time of delivery could also be associated with a fatal outcome alongside birth weight, irrespective of the lack of significance in our results. A similar explanation was proposed in a prospective cohort study of 172 newborns with sepsis. The newborns with extremely low birth weight, defined as a weight of 1000 g, had the highest mortality rate (65.7%) compared to other birth-weight categories of patients [27]. This population born before the 29th week of gestation also had a higher risk of the development of other conditions already proven to be significantly associated with a higher mortality rate [28], which could contribute to the increased frequency of fatal outcomes in our case group of patients. According to our results, a 1 g increase in birth weight would decrease sepsis-related death risk by 0.2%. 

Recently, the requirement for inotropic support was assessed to predict the mortality rate in newborns with sepsis. In a study population similar to ours, weighing under 1500 g and born before 32 weeks of gestation, the mortality risk was shown to be 22 times higher in a group of newborns requiring inotropic support [29]. The pooled OR for the need for vasoactive agents obtained from 20 studies analyzed in a recently published meta-analysis was 6.5 [23]. It has been reported that newborns with sepsis who are at high risk of death develop multiple cardiovascular, respiratory, immune, and renal failures [30], and that the need for inotropic drugs to support the cardiovascular system is significantly increased in this group of patients [31]. The development of such multiorgan failure can be utilized to predict newborn death rates. The Neonatal Sequential Organ Failure Assessment Score (nSOFA) is a recently introduced tool for the evaluation of the presence and progression of specific signs of multiorgan failure. One of its determinants for the evaluation of the cardiovascular system is, precisely, the utilization of two or more inotropes. The mortality rate was significantly higher in the group of newborns with an nSOFA score >4 [31]. Our results also support the association between double-inotropic treatment and the risk of a fatal outcome, suggesting the importance of the timely recognition of multiorgan dysfunction to reduce sepsis mortality. The nSOFA scoring system or some other tool for organ failure assessment is needed in routine practice.

The progression to dysfunction of different organs due to sepsis, lower oxygen delivery, and a rapid decline in circulating red blood cells (RBC) and hemoglobin are some of the indications for transfusions in preterm newborns to prevent shock and a fatal outcome [32]. More than 90% of preterm infants are likely to develop anemia of prematurity [33], which is significantly associated with a fatal outcome if sepsis is present [27]. Furthermore, very low-birth-weight preterm infants with sepsis were shown to be 3.22 times more likely to receive RBC transfusions [34]. Nearly 90% of newborns who weigh less than 1000 g may require an average of five RBC transfusions while they are in the hospital [35]. In addition to sepsis, our patients in the case group had considerably lower hemoglobin levels than those in the control group, which may have been the main factor in the need for RBC transfusions. However, although with the aim to prevent the poor outcomes of sepsis, erythrocyte transfusion is also associated with the development of life-threatening conditions including necrotizing enterocolitis, bronchopulmonary dysplasia, and intraventricular hemorrhage that could diminish the benefits of its utilization [33]. The same authors found that the mortality rate among preterm newborns receiving erythrocytes was 50% higher during the first 28 days of life. The results of another study showed an association between a fatal outcome after 28 days of life and the number of transfusions. When comparing newborns who received three or more erythrocyte transfusions to those who received fewer than three, the relative risk of death was 89% higher [34]. It is possible that our patients developed some of the aforementioned conditions, which increased the rate of fatal outcomes compared to the control group. The other reasons for adverse conditions after erythrocyte transfusion may be related to the donor, the time of the erythrocyte bioactive substance accumulation, or artificial additives [33]. However, there are studies that reported no difference in the incidence of mortality or multiple-organ dysfunction in newborns who received fresh blood cells compared to those who received the oldest blood units available [32]. Given the inconsistency, future studies with sufficient power are needed to confirm the association between erythrocyte transfusion and poor outcomes from sepsis in newborns.

In our study, the most common cause of sepsis was GNB, with *Klebsiella*, *Acinetobacter*, *Pseudomonas*, *Escherichia coli*, *Serratia*, and *Citrobacter* being most frequently identified. This conclusion is in line with the research-based data. Among hospitalized neonates who are more likely to experience septic shock and die, the rate of GNB sepsis rises. In this case-control study analyzing patients’ data over 27 years, the frequency of GNB sepsis was 66%, and a fatal outcome was observed in 33% of newborns compared to 20% in newborns with non-GNB sepsis. The main reasons for increased death in the case group were a higher risk of septic shock and a lack of adequate medication [36]. Similarly, GNB sepsis ended fatally in 67.7% (*Acinetobacter*), 58.2% (*Klebsiella*), and 58.3% (*Pseudomonas*) of patients, with multiple antibiotic resistance as the only confirmed predictor of fatal outcome [37]. The main contributors to a fatal outcome in newborns with *Acinetobacter* sepsis were septic shock, with odds of 41.38, and inadequate antibiotic treatment, showing a 10-times-higher risk [38]. According to a recent systematic review, antibiotic resistance caused by insufficient empiric care is a serious concern in the rising prevalence of GNB sepsis in infants [39]. It is possible that empiric antibiotic treatment, multidrug resistance, and the development of septic shock in our population are the reasons for the sevenfold higher likelihood of a fatal outcome in newborns with GNB sepsis. 

Our analysis was limited by the data being collected from a single institution, and the ratio between the groups. Nevertheless, we managed to include a total of 136 newborns over a period of 3 years, a ratio between groups of almost 1:2, which we consider sufficient data for analysis and valid conclusions. The absence of data on antibiotic resistance, which could cast additional light on the connection between GNB sepsis and mortality rates, is another shortcoming.

## 5. Conclusions

The fatal outcome of sepsis in newborns was associated with birth weight, double-inotropic treatment, gram-negative bacteria, and erythrocyte transfusion. Our results also revealed the importance of additional tools for timely diagnosis, organ failure assessment, and the appropriate guidance for treatment options that may decrease multidrug resistance and the mortality rate of sepsis in newborns.

## Figures and Tables

**Table 1 medicina-59-00604-t001:** The basic clinical characteristics of the pregnant women.

Variable	Mean ± Standard Deviation (Range) or Number (%)
Conception	
Natural	114 (85.7%)
Assisted reproduction	19 (14.3%)
Number of fetuses	
Singleton pregnancy	96 (72.2%)
Multiple pregnancy	37 (27.8%)
Number of previous births: 0/1/2/3/4/5	59 (42.7%)/51 (38.3%)/13 (9.8%)/5 (3.8%)/4 (3.0%)/1 (0.7%)
History of miscarriages	27 (20.3%)
Threatened premature labor	36 (27.1%)
Premature rupture of membranes	33 (24.8%)
Amniotic fluid pathology	26 (19.5%)
Acute and chronic diseases	
Urogenital infection/chorioamnionitis	39 (28.3%)
Hypertension	24 (18.0%)
Diabetes mellitus	5 (3.8%)
Any smoking during pregnancy	12 (9.0%)

**Table 2 medicina-59-00604-t002:** The basic characteristics of the newborns at birth.

Variable	Cases (*n* = 51) Mean ± Standard Deviation (Range) or Number (%)	Controls (*n* = 85) Mean ± Standard Deviation (Range) or Number (%)	Test Value and *p* Value
Gender			
Female	23 (45.1%)	34 (40.0%)	χ^2^ = 0.163, *p* = 0.686
Male	28 (54.9%)	51 (60.0%)	
Birth weight (g)	882.8 ± 372.2 (400–2850)	1660.9 ± 721.1 (650–3600)	U = 507.500, *p* = 0.000 *
Time of delivery ^1^			
Extremely preterm	38 (74.5%)	19 (22.4%)	χ^2^ = 43.368, *p* = 0.000 *
Very preterm	12 (23.5%)	26 (30.6%)	
Moderate preterm	0 (0%)	15 (17.6%)	
Late preterm	0 (0%)	18 (21.2%)	
Early term	1 (2%)	5 (5.9%)	
Full term	0 (0%)	2 (2.4%)	
Delivery method			
Vaginal	27 (53%)	38 (44.7%)	χ^2^ = 1.993, *p* = 0.369
Caesarean section	24 (47.1%)	47 (55.3%)	
Developmental level ^2^			
SGA	7 (13.7%)	12 (14.1%)	χ^2^ = 0.867, *p* = 0.285
AGA	43 (84.3%)	70 (82.4%)	
LGA	1 (2.0%)	3 (3.5%)	
Ponderal index	2.2 ± 0.4 (1.2–3.3)	2.2 ± 0.3 (1.3–3.0)	U = 2109.500, *p* = 0.794
Apgar score at first minute			
8–10	1 (2.0%)	21 (24.7%)	χ^2^ = 24.288, *p* = 0.000 *
4–7	16 (31.4%)	41 (48.2%)	
0–3	34 (66.7%)	23 (27.1%)	
ABO blood group type			
A	23 (46.0%)	34 (40.0%)	χ^2^ = 1.383, *p* = 0.710
B	8 (16.0%)	11 (12.9%)	
AB	4 (8.0%)	6 (7.1%)	
O	15 (30.0%)	34 (40.0%)	
Rh blood group type			
Rh−	8 (15.7%)	11 (12.9%)	χ^2^ = 0.042, *p* = 0.622
Rh+	43 (84.3%)	74 (87.1%)	

* Statistically significant. ^1^ Extremely preterm—<28 weeks; very preterm—28 0/7 to 31 6/7 weeks; moderate preterm—32 0/7 to 33 6/7 weeks; late preterm—34 0/7 to 36 6/7 weeks; early term—37 0/7 to 38 6/7 weeks; full term—39 0/7 to 40 6/7 weeks. ^2^ SGA—small for gestational age; AGA—appropriate for gestational age; LGA—large for gestational age.

**Table 3 medicina-59-00604-t003:** The biochemical and clinical characteristics of the newborns.

Variable	Cases (*n* = 51) Mean ± Standard Deviation (Range) or Number (%)	Controls (*n* = 85) Mean ± Standard Deviation (Range) or Number (%)	Test Value and *p* Value
Cause of sepsis			
Gram-negative bacteria ^1^	47 (92.5%)	45 (52.9%)	χ^2^ = 20.642, *p* = 0.000 *
Gram-positive bacteria ^2^	4 (7.8%)	40 (47.1%)	
Average value of blood pH on the 1st day	7.26 ± 0.11 (6.86–7.48)	7.32 ± 0.08 (7.07–7.46)	U = 1407.500, *p* = 0.001 *
Average value of base excess on the 1st day (mmol/L)	−6.35 ± 3.78 (−21.2–2.67)	−4.64 ± 2.73 (−12.0–1.57)	U = 1494.500, *p* = 0.002 *
Body temperature on admission (°C)	35.52 ± 0.80 (32.90–36.90)	36.15 ± 0.54 (34.70–37.30)	U = 1035.500, *p* = 0.000 *
Average value of FiO_2_ ^3^ during the 1st day	49.7 ± 15.6 (27.0–100.0)	44.6 ± 13.8 (21.0–85.0)	U = 1745.500, *p* = 0.058
Average value of systolic blood pressure on the 1st day (mmHg)	51.84 ± 10.87 (26.00–83.00)	59.06 ± 11.71 (42.00–92.00)	U = 1335.000, *p* = 0.000 *
Average value of diastolic blood pressure on the 1st day (mmHg)	25.61 ± 9.46 (8.00–59.00)	31.24 ± 8.68 (16.00–60.00)	U = 1278.500, *p* = 0.000 *
Average value of mean blood pressure on the 1st day (mmHg)	36.87 ± 9.65 (15.00–65.00)	43.80 ± 9.91 (27.00–81.00)	U = 1222.500, *p* = 0.000 *
Leukocytes on the 1st day (×10^9^/L)	19.36 ± 10.96 (4.30–68.90)	20.42 ± 13.56 (3.60–94.60)	U = 2107.500, *p* = 0.787
Thrombocytes on the 1st day (×10^9^/L)	181.78 ± 59.74 (49.00–310.00)	209.05 ± 61.45 (88.00–348.00)	U = 1694.500, *p* = 0.021 *
Hemoglobin on the 1st day (g/L)	169.86 ± 29.80 (88.00–221.00)	182.33 ± 28.81 (86.00–250.00)	U = 1684.000, *p* = 0.030 *
Albumin on the 1st day (g/L)	25.82 ± 5.03 (12.00–42.00)	30.51 ± 5.08 (19.00–42.00)	U = 1026.500, *p* = 0.000 *
Urea on the 3rd day (mmol/L)	9.73 ± 2.49 (5.20–16.80)	6.43 ± 3.61 (1.20–18.9)	U = 668.500, *p* = 0.000 *
Creatinin on the 3rd day (mmol/L)	95.41 ± 25.91 (49.00–166.00)	74.89 ± 21.63 (23.00–161.00)	U = 842.000, *p* = 0.000 *
Bilirubin on the 2nd day (μmol/L)	76.91 ± 28.29 (17.00–139.00)	88.97 ± 34.51 (27.00–229.00)	U = 1764.000, *p* = 0.100
Convulsions during 1st week			
Yes	33 (64.7%)	18 (21.2%)	χ^2^ = 23.946, *p* = 0.000 *
No	18 (35.35)	67 (78.8%)	
Pneumothorax/Pneumomediastinum/Pulmonary interstitial emphysema			
Yes	10 (19.6%)	7 (8.2%)	χ^2^ = 3.132, *p* = 0.077
No	41 (80.4%)	78 (91.8%)	

* Statistically significant. ^1^
*Klebsiella*, *Acinetobacter*, *Pseudomonas*, *Escherichia coli*, *Serratia*, *Citrobacter*. ^2^
*Staphylococcus*, *Enterococcus*. ^3^ The fraction of inspired oxygen.

**Table 4 medicina-59-00604-t004:** Conducted therapeutic measures in newborns.

Variable	Cases (*n* = 51) Mean ± Standard Deviation (Range) or Number (%)	Controls (*n* = 85) Mean ± Standard Deviation (Range) or Number (%)	Test Value and *p* Value
Resuscitation measures			
Yes	40 (78.4%)	47 (55.3%)	χ^2^ = 17.047, *p* = 0.001 *
No	11 (21.6%)	38 (44.7%)	
Ibuprofen for the treatment of ductus arteriousus			
Yes	16 (31.4%)	19 (22.4%)	χ^2^ = 0.926, *p* = 0.336
No	35 (68.6%)	66 (77.6%)	
Surfactant			
Yes	22 (43.1%)	20 (23.5%)	χ^2^ = 4.859, *p* = 0.027 *
No	29 (56.9%)	65 (76.5%)	
Double-inotropic therapy			
Yes	41 (80.4%)	17 (20.0%)	χ^2^ = 45.093, *p* = 0.000 *
No	10 (19.6%)	68 (80.0%)	
Phototherapy			
Yes	41 (80.4%)	76 (89.4%)	χ^2^ = 1.405, *p* = 0.236
No	10 (19.6%)	9 (10.6%)	
Initial antibiotic therapy			
Ampicillin + Gentamycin	11 (21.6%)	27 (31.8%)	χ^2^ = 2.091, *p* = 0.352
Ampicillin + Amikacin	39 (76.5%)	55 (64.7%)	
Ampicillin + Meropenem	1 (2.0%)	3 (3.5%)	
Protein intake during the 1st day (g/kg)	0.5 ± 0.6 (0–3.3)	0.4 ± 0.5 (0–2.0)	U = 1871.000, *p* = 0.160
Glucose intake during the 1st day (mg/kg/min)	5.1 ± 1.3 (1.0–8.7)	5.0 ± 1.2 (0.8–8.4)	U = 2148, *p* = 0.930
Fluid intake during the 1st day (mL/kg)	99.0 ± 17.4 (58.0–156.0)	85.7 ± 17.1 (13.0–139.0)	U = 1142.500, *p* = 0.000 *
Erythrocyte transfusion during the 1st week			
Yes	47 (92.2%)	22 (25.9%)	χ^2^ = 53.394, *p* = 0.000 *
No	4 (7.8%)	63 (74.1%)	
Thrombocyte transfusion during the 1st week			
Yes	29 (56.9%)	10 (11.8%)	χ^2^ = 29.530, *p* = 0.000 *
No	22 (43.1%)	75 (88.2%)	
Plasma transfusion during the 1st week			
Yes	40 (78.4%)	33 (38.8%)	χ^2^ = 18.549, *p* = 0.000 *
No	11 (21.6%)	52 (61.2%)	

* Statistically significant.

**Table 5 medicina-59-00604-t005:** Crude and adjusted odds ratios (OR) of the risk factors for fatal outcome in early neonatal sepsis.

Risk Factors	Univariate Model Crude OR with 95% CI *p*	Multivariate Model Adjusted # OR with 95% CI *p*
Birth weight	0.996 (0.995–0.998) *p* = 0.000 *	0.998 (0.996–1.000) *p* = 0.046 *
Average value of blood pH on the 1st day	0.001 (0.000–0.067) *p* = 0.001 *	0.010 (0.000–3.334) *p* = 0.121
Body temperature on admission	0.212 (0.107–0.421) *p* = 0.000 *	0.641 (0.265–1.552) *p* = 0.324
Double-inotropic therapy	16.400 (6.857–39.222) *p* = 0.000 *	9.186 (2.451–34.432) *p* = 0.001 *
Convulsions during 1st week	6.824 (3.144–14.812) *p* = 0.000 *	2.126 (0.621–7.283) *p* = 0.230
Erythrocyte transfusion during the 1st week	33.648 (10.866–104.198) *p* = 0.000 *	5.279 (1.147–24.290) *p* = 0.033 *
Plasma transfusion during the 1st week	5.730 (2.582–12.717) *p* = 0.000 *	1.290 (0.320–5.198) *p* = 0.721
Blood culture	10.444 (3.455–31.569) *p* = 0.000 *	7.071 (1.147–43.584) *p* = 0.035 *

*p*: statistical significance; CI: confidence interval; *: statistically significant; #: adjusted for average value of blood pH on the first day, body temperature on admission, convulsions during first week, and plasma transfusion during the first week.

## Data Availability

Data are available upon request from corresponding authors.

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
