# Peer review of "Predictors of Mortality in Early Neonatal Sepsis: A Single-Center Experience"

_medicina, 2023, doi:10.3390/medicina59030604_

Round 1
Reviewer 1 Report
This single centre, case control, retrospective study investigated factors that predicted mortality in early onset neonatal sepsis. The authors identified low birth weight, presence gram negative sepsis, dual inotrope use, and red blood cell transfusion as predictors for mortality.
Abstract:
1. Consider stating the birth weight and gestation studied rather than maternal age in the abstract
Introduction:
1. Overall, this is well written.
2. Line 53: The readers get the impression that procalcitonin……. IL-8 were examined in this study. Kindly rephrase
Methods:
1. Were the controls matched for gestation or birth weight? If not, why not.
2. Line 80 and – typo – stating up to 2025. Please correct
3. Line 83 – typo.
4. Line 124 – last sentence ending 95%. Is there a missing word after this?
Results:
1. Line 127; consider stating “women with mean (SD) of ….”
2. Please tabulate lines 128 to 140. It is very difficult to read this text
3. Line 157 to 175 – please present this data in table 1 and 2. It is much easier to follow this data in a tabular form rather than in text. Also indicate the significant parameters in the table.
4. Line 180-190 – please indicate in tabular form
5. Table 1 – 3 – indicated significant values in the tables. You can remove the column stating total from all the tables. This does not add any further information and makes the tables look too busy.
Discussion
Line 290 – typo
Author Response
University of Kragujevac
Faculty of Medical Sciences
10.03.2023.
To: Editors of Medicina
Dear Editors,
we have made revision of our article 2274464 entitled “Predictors of mortality in early neonatal sepsis: a single-center experience” according to the requests of the reviewers. Changes in the manuscript are shown in track changes.
Reviewer 1
Comment 1: Abstract:
- Consider stating the birth weight and gestation studied rather than maternal age in the abstract
Author’s response:
Thank you for this advice. In the abstract of the revised manuscript, we omitted the average age of the pregnant women and provided data on the birth weight of the newborns and the gestational age of the newborns at delivery by group.
Comment 2: Introduction:
- Overall, this is well written.
Author’s response:
Thank you for this comment and for taking the time to review our work.
Comment 3: Introduction:
- Line 53: The readers get the impression that procalcitonin……. IL-8 were examined in this study. Kindly rephrase.
Author’s response:
We are grateful to the reviewer for the indicated chance to improve the quality of our article. We rephrased this sentence. The rephrased sentence reads: “Hence, clinicians are continually searching for new markers that might provide high selectivity and specificity for the diagnosis of early neonatal sepsis, such as inflammatory markers, cytokines, different hematological parameters, and indicators for determining nutritional status [14-21].”
Comment 4: Methods:
- Were the controls matched for gestation or birth weight? If not, why not.
Author’s response:
We are particularly grateful to the reviewer for this comment. We misrepresented the study design in the material and method section. Our study is not a case-control study but a retrospective cross-sectional study. That is why we did not match the respondents according to birth weight or gestational age. Instead, we considered birth weight and gestational age as confounding variables, whose effect on outcome we also studied. We corrected the study design and noted that it was a retrospective cross-sectional study in the abstract and materials and methods of the revised manuscript.
Comment 5: Methods:
- Line 80 and – typo – stating up to 2025. Please correct.
Author’s response:
Thank you for this remark. We corrected “2025” to “2015” in the revised version of the manuscript.
Comment 6: Methods:
- Line 83 – typo.
Author’s response:
Thank you for this remark. We corrected that technical error in the revised version of the manuscript.
Comment 7: Methods:
- Line 124 – last sentence ending 95%. Is there a missing word after this?
Author’s response:
Thank you for this remark. You are right; the phrase "confidence interval" is missing after the word 95. We corrected that in the revised version of the manuscript.
Comment 8: Results:
- Line 127; consider stating “women with mean (SD) of ….”
Author’s response:
Thank you for the suggestion, which we have accepted. We rephrased that sentence in the revised version of the manuscript in the manner suggested. The rephrased sentence reads: “The study included 133 pregnant women with a mean (SD) age of 30.5 (5.7) years.”
Comment 9: Results:
- Please tabulate lines 128 to 140. It is very difficult to read this text.
Author’s response:
We are grateful to the reviewer for the indicated chance to improve the quality of our article. We relocated the required data that was previously listed in text into a table entitled: “Table 1. The basic clinical characteristics of pregnant women”.
Comment 10: Results:
- Line 157 to 175 – please present this data in table 1 and 2. It is much easier to follow this data in a tabular form rather than in text. Also indicate the significant parameters in the table.
Author’s response:
Once again we are grateful to the reviewer for the indicated chance to improve the quality of our article. We presented the required data in tables in the revised version of the manuscript. Also, we changed the tables by inserting data with statistical test results and p values rather than a column with total data in accordance with one of the reviewer's subsequent comments (“Table 1 – 3 – indicated significant values in the tables. You can remove the column stating total from all the tables. This does not add any further information and makes the tables look too busy”). Because we added Table 1 with the clinical data of pregnant women, there was a shift in the number under which the tables are listed (Table 1 from the primary version of the manuscript is labeled "Table 2," and so on). We indicated all significant parameters (p<0.05) with asterisk.
Comment 11: Results:
- Line 180-190 – please indicate in tabular form
Author’s response:
Thank you for this remark. We presented the required data in Table 4 entitled “Conducted therapeutic measures in newborns.”
Comment 12: Results:
- Table 1 – 3 – indicated significant values in the tables. You can remove the column stating total from all the tables. This does not add any further information and makes the tables look too busy.
Author’s response:
We are particularly grateful to the reviewer for this comment. As we previously mentioned, we removed the column stating total from all the tables. Instead of this data, we stated the data with the values of statistical tests and p values with an indication of the significant parameters.
Comment 13: Discusion:
- Line 290 – typo.
Author’s response:
Thank you for this remark. We corrected that technical error in the revised version of the manuscript.

Reviewer 2 Report
The case group has a lower birth weight and fetal age, and this factor itself can be the main reason for more infant mortality. What is the reason that this confounding variable is not the same in two groups?
The impact of many variables mentioned in the study on the mortality rate of infants with sepsis has been investigated. What is the innovation of your study?
Author Response
University of Kragujevac
Faculty of Medical Sciences
10.03.2023.
To: Editors of Medicina
Dear Editors,
we have made revision of our article 2274464 entitled “Predictors of mortality in early neonatal sepsis: a single-center experience” according to the requests of the reviewers. Changes in the manuscript are shown in track changes.
Reviewer 2:
Comment 1:
The case group has a lower birth weight and fetal age, and this factor itself can be the main reason for more infant mortality. What is the reason that this confounding variable is not the same in two groups?
Author’s response:
We are particularly grateful to the reviewer for this comment. We misrepresented the study design in the material and method section. Our study is not a case-control study but a retrospective cross-sectional study. That is why we did not match the respondents according to birth weight or gestational age. Instead, we considered birth weight and gestational age as confounding variables, whose effect on outcome we also studied. We corrected the study design and noted that it was a retrospective cross-sectional study in the abstract and materials and methods of the revised manuscript.
Comment 2:
The impact of many variables mentioned in the study on the mortality rate of infants with sepsis has been investigated. What is the innovation of your study?
Author’s response:
We agree with the reviewer that the impact of many variables from our study on the mortality rate of infants with sepsis has already been investigated. However, it is important to note that there are significant controversies regarding the degree and direction of influence of individual variables on the outcome of interest. The results of our study might be used to explain the direction and strength of some predictors of mortality in early neonatal sepsis. This, in our opinion, is what gives our work its originality and clinical relevance.